
# A wave-resolving 2DV Lagrangian approach to model microplastic transport in the nearshore

Isabel Jalón-Rojas[1], Damien Sous[2,3], and Vincent Marieu[1]

[1]Univ. Bordeaux, CNRS, Bordeaux INP, EPOC, UMR 5805, F-33600 Pessac, France
[2]Université de Toulon, Aix Marseille Université, CNRS, IRD, Mediterranean Institute of Oceanography (MIO), La Garde, France
[3]Université de Pau et des Pays de l'Adour, E2S UPPA, SIAME, Anglet, France

**Correspondence:** Isabel Jalón-Rojas (isabel.jalon-rojas@u-bordeaux.fr)

**Abstract.** Potentially acting as a source or a sink for plastic pollution to the open ocean, nearshore waters remain a challenging context for predicting the transport and deposition of plastic debris. In this study, we present an advanced modelling approach based on the SWASH wave model and the TrackMPD (v3.0) particle transport model to investigate the transport dynamics of floating and sinking microplastics in wave-dominated environments. This approach introduces novel features such as coupling

with advanced turbulence models, simulating resuspension and bedload processes, implementing advanced settling and rising velocity formulations, and enabling parallel computation. The wave laboratory experiments conducted by Forsberg et al. (2020) were simulated to validate the model's ability to reproduce the transport of diverse microplastics (varying in density, shape, and size) along a comprehensive beach profile, capturing the whole water column. Our results underscore the robustness of the proposed model, showing good agreement with experimental data. High-density microplastics moved onshore near the bed

accumulating in proximity to the wave-breaking zone, while the distribution of low-density particles varied along the coastal profile depending on the particle properties. The study also sheds light on the primary mechanisms driving microplastic transport, such as Stokes drift, wave asymmetry and settling/rising velocities. Sensitivity analyses on calibration parameters further confirm the robustness of the model results and the influence of these factors on transport patterns. This research establishes the SWASH-TrackMPD approach as a valuable tool, opening avenues for future studies to contextualize laboratory findings within

the complexities of real-world nearshore environments and further refine our comprehension of microplastic dynamics across different beaches and wave-climate conditions.

## 1 Introduction

Nearshore areas are highly dynamic environments influenced by complex interactions among water, sediment, biota, and human activity. These regions support high productivity, biodiversity, and provide multiple ecosystem services (Liu and Stern, 2008;

McLachlan and Defeo, 2017). However, they are also vulnerable to anthropogenic and natural threats, such as sea-level rise, extreme storms and pollution (Rippy et al., 2013; d'Anna et al., 2022). Microplastic (MP, 0.1 $\mu$m to 5 mm plastic particles) pollution has emerged as a major concern, with the potential to cause significant ecological, aesthetic, and economic impacts on beaches and coastal environments. Numerous studies have reported MP pollution on beaches globally, with some revealing





alarmingly high contamination levels (Turra et al., 2014; Fok and Cheung, 2015; Pérez-Alvelo et al., 2021; Tata et al., 2020;
Lefebvre et al., 2021).

Despite the well-documented presence and abundance of MPs on beaches, fundamental questions regarding their transport,
dispersion, trapping and fate in nearshore waters remain unresolved (Zhang, 2017). This is primarily due to (1) the intricate
nature of nearshore hydrodynamic processes and the rapid morphodynamic changes occurring within these environments
(Castelle and Masselink, 2023); (2) the inherent variability of MP properties, including density, size and shape, as well as
complex processes like biofouling, aggregation, and fragmentation, which influence their transport behaviour (Khatmullina and
Chubarenko, 2019; Jalón-Rojas et al., 2022); (3) the multiple pathways through which MPs enter the nearshore environment,
including runoff from urban areas and rivers, direct discharge from ships or offshore sources, and beach litter (Lefebvre et al.,
2023); and (4) the lack of high-resolution data and suitable modelling approaches to accurately capture the multidimensional
aspects of MP transport in these environments.

The absence of direct field observations beyond the beach region (Chubarenko et al., 2018), within or near the surf zone water
column, can be primarily attributed to the inherent difficulties associated with sampling MPs in this dynamic and often hazardous
environment. Consequently, previous studies investigating microplastic transport in nearshore waters have predominantly
relied on laboratory experiments in wave or wind-wave flumes (Forsberg et al., 2020; Kerpen et al., 2020; Guler et al., 2022;
Larsen et al., 2023; Núñez et al., 2023). While these recent experiments have provided valuable insights into certain aspects
of microplastic movement, they remain small-scale idealized representations of nearshore systems, affected by scaling issues.
Furthermore, laboratory experiments on wave- or wind-wave-driven transport of MP are very costly approaches, which limits
the number of tested configurations and their ability to capture the diversity and complexities of real-world scenarios.

To overcome these limitations, numerical modeling offers a promising tool to elucidate the transport and fate of microplastics
in nearshore waters, providing a means to explore multiple scenarios that may be challenging to investigate through field or
laboratory experiments (e.g., different forcings, beach morphologies, and particle characteristics). Although numerical models
have been extensively developed at ocean and regional scales in recent years (e.g. Dobler et al., 2019; Jalón-Rojas et al., 2019;
Lobelle et al., 2021; Baudena et al., 2022), there is a significant knowledge gap in modelling MP dynamics at the nearshore scale.
In particular, there is a pressing need to develop numerical tools able to address the wave- and particle-resolved time and space
scales of transport. Such a detailed characterization of MP dynamics remains a necessary step to avoid the use of uncontrolled
bulk parameterization of transport processes. To our knowledge, prior to this study, Stocchino et al. (2019) have conducted such
a fine numerical study evaluating the effects of sea waves on inertial microplastic dynamics. However, this study was mainly
dedicated to deep water transport and several hydrodynamic processes affecting the transport in nearshore areas, such as wave
asymmetry and wave breaking, and transport mechanisms like resuspension or bedload, were not considered. Additionally, a
more recent study by Kim and Kim (2024) modeled microplastic transport in the nearshore region, emphasizing the key role of
wave breaking and rip currents. However, it exclusively focused on buoyant particles and did not consider near-bed transport
processes such as deposition, resuspension, and bedload.

In this study, we aim to develop a wave-resolving 2D lagrangian numerical model based on SWASH (Zijlema et al., 2011) and
TrackMPD (Jalón-Rojas et al., 2019) models to simulate the transport of floating and sinking microplastics (MPs) in shallow,





wave-dominated environments. A major challenge for process-based modeling of the complex nearshore microplastic transport
is the lack of field observations. In this context, data from laboratory studies offer a valuable opportunity to validate numerical
model approaches. We use wave laboratory experiments conducted by Forsberg et al. (2020) as a validation benchmark to assess
the model's capacity in simulating the transport of different types of microplastics (different shapes, densities and sizes) along a
2D beach profile by describing the entire water column. Furthermore, we evaluate the sensitivity of model parameters to assess
their influence on the simulated transport trends.

## 2   Methods

The process-based hydrodynamical model SWASH was utilized in this study to simulate the wave-driven hydrodynamics and
generate the associated current velocity field, which is subsequently provided (offline) as input to the lagrangian particle-tracking
model TrackMPD. In this section, we provide a concise overview of both models and present the novel developments made in
this study to enhance the modeling of key transport mechanisms in nearshore waters. The test case from Forsberg et al. (2020)
and the sensitivity scenarios are then detailed.

### 2.1   The base models

#### 2.1.1   SWASH

SWASH (Zijlema et al., 2011) is a non-hydrostatic, wave-resolving model designed to solve the nonlinear shallow water
equations derived from the incompressible Navier-Stokes equations. SWASH has been particularly designed to simulate the
transformation of surface waves as they propagate toward the shore, capturing the key features of surf and swash zone dynamics
including nonlinear shoaling, wave breaking, wave runup, and wave-driven currents. The model employs an explicit, second-
order finite difference method that ensures the conservation of both mass and momentum at the numerical level. In the present
study, unidirectional cross-shore propagating waves were imposed at the left boundary of the domain, so the equations are solved
in a two-dimensional vertical (2DV) plane. A structured grid is utilized for discretizing the physical domain, with a constant
width in the x-direction and a fixed number of layers between the bottom and the free surface in the vertical direction. The
layer thickness is defined as a constant fraction of the water depth, similar to the sigma-layer coordinate system. SWASH has
been validated and applied extensively in laboratory and field studies to investigate nearshore waves and wave-driven processes
(Zijlema et al., 2011; Rijnsdorp et al., 2014; Suzuki et al., 2017; Zhang et al., 2018; Sous et al., 2020). Further details on the
present SWASH configuration are given in Section 2.3.

#### 2.1.2   TrackMPD

TrackMPD (Jalón-Rojas et al., 2019) is a lagrangian particle-tracking model that incorporates a wide range of particle transport
processes, including advection, turbulent dispersion, beaching, refloating, deposition, and resuspension. It also accounts for
various microplastic behaviours such as settling/rising, biofouling, and degradation, which are contingent upon the physical



properties of the particles. The model employs a $4^{th}$-order Runge-Kutta scheme to accurately advect virtual particles through a
set of velocity fields. A random-walk approach is implemented to simulate the turbulent motion of particles in both the horizontal
and vertical directions as a function of the horizontal and vertical diffusivity coefficients (see Section 2.2.1 for more details). The
settling or rising velocities of particles can be defined by users or calculated online using various empirical formulations that
account for particle physical properties and other behaviors such as biofouling and degradation. TrackMPD is an open-source
model and can be accessed on GitHub (https://github.com/IJalonRojas/TrackMPD). For more information about TrackMPD v1,
readers can refer to (Jalón-Rojas et al., 2019).

## 2.2 Developments

In order to enhance the Lagrangian modelling of particles in nearshore and coastal waters, significant advancements have
been incorporated into TrackMPD. The focus is placed on the dynamical representation of two strong controlling factors of
nearshore transport: (i) turbulent mixing, which displays striking spatial gradients forced by wave breaking and depth variations
and, (ii), the main time-varying processes governing the transport of solid particles in response to the continuously evolving
hydrodynamical conditions (water depth, current, shear stress), particularly deposition, erodability, and subsequent transport.
These developments ultimately led to the release of TrackMPD v2.3 and v3.0 (used in this work), which offer several key
features and improvements compared to the previous version.

Firstly, TrackMPD (from v2.3) allows for the off-line coupling of turbulence models, as detailed in Section 2.2.1, enabling
a more accurate representation of turbulent dispersion effects on particle transport. Secondly, additional processes such as
deposition, resuspension, and bedload have been included or improved, providing a more accurate representation of particle
dynamics (Sections 2.2.2 and 2.2.3). The resuspension process was built upon the initial simplified proposition presented by
(Cheng et al., 2020). Furthermore, parallel computation of particle trajectories has been implemented, leading to faster simulation
times and improved computational efficiency. Additionally, while in the first version, advection was computed at an internal
time step, and dispersion and behaviour were calculated at the output time step, the new version integrates these processes
into a unified internal time step of calculation. New parameterizations for settling and rising velocities (Dellino et al., 2005;
Waldschläger and Schïtrumpf, 2019a; Jalón-Rojas et al., 2022) have been also introduced. Finally, TrackMPD includes now a
verbose mode that allows users to select the degree of detail in the progress messages during simulation runs.

It is important to note that TrackMPD retains its availability for both the 2DH and 3D approaches. However, for the specific
focus of this work, we have adapted the 3D approach to a 2DV approach, focusing specifically on the dynamics within a
two-dimensional vertical plane.

### 2.2.1 Vertical Diffusivity

TrackMPD (from v2.3) offers the flexibility to set the horizontal and vertical diffusivity coefficients either as constant or varying
values. In the latter case, the diffusivity coefficients are generated by an external hydrodynamic or turbulence model and read by
TrackMPD in a similar way to current velocities. This approach enables the diffusivity coefficients to vary over space and time,
providing a more realistic representation of the particles' transport processes.





In this study, the vertical turbulent "eddy" diffusivity $\nu_t$ was estimated from the non-dimensional analysis of surf zone turbulence presented by Feddersen (2012). This analysis has demonstrated to provide consistent scaling for both laboratory and natural surf zone turbulence. A spectral analysis was first performed on each SWASH wave run to extract the energy flux

$F$ [N.s$^{-1}$] over the whole domain based on the linear theory. A wave-averaged vertical eddy diffusivity [m$^2$/s] is therefore estimated as:

$$\nu_t(x,z) = Ah(\frac{1}{\rho}\frac{dF}{dx})^{1/3}exp(B\frac{z}{h}) \qquad (1)$$

where $x$ and $z$ and the horizontal cross-shore and vertical coordinates, $h$ is the local depth [m], $\rho$ is the density of water [kg.m$^3$], $A = 0.0147$ and $B = 1.46$ following the coefficient values proposed by Feddersen (2012). The wave-averaging means that the

wave-breaking-induced eddy diffusivity is supposed to be driven by the turbulent field resulting from the breaking of many successive waves (Ruessink, 2010). For the present simulation, an additional background eddy viscosity is imposed in two zones. First, a constant value is imposed in the beach zone for which the Feddersen scaling is not applicable due to the impossibility to compute energy flux in the intermittently wet and dry portion of the beach. Second, offshore the surfzone, eddy viscosity is introduced to account for turbulence generated in the vicinity of the wave maker. A smoothing is applied to avoid sharp $\nu_t$

gradients in transition areas.

The vertical diffusivity coefficient for particles, $K_v$, is here assumed to be equal to the eddy diffusivity, i.e. the assumption is made that momentum and plastic particles diffuse at the same rate. The horizontal diffusivity coefficient for particles is here kept uniform throughout the domain.

### 2.2.2 Deposition, resuspension and bedload conditions

Depending on user preferences, different types of particle interactions with the bed can be set. Particles reaching the bed can be deposited. Once at the bed particles can whether be temporarily or permanently attached to the bed or left free for further motions. Particles can be considered as definitely trapped in the bed if strong interactions with bed sediments are expected. It is also possible to dynamically evaluate their fate at each time step according to the prevailing hydrodynamic conditions. In the latter case, the particle shear stress [N/m$^2$] is estimated at the position of the deposited particle at each time step as:

$$\tau_0 = \rho\nu_t \frac{\partial U}{\partial z}|_{z=0} \qquad (2)$$

where $\rho$ is the density of water. The shear stress can be approximated as:

$$\tau_0 = \rho\nu_t \frac{U_1}{\Delta z} \qquad (3)$$

where $U_1$ [m/s] and $\Delta z$ [m] are the first (bottom) layer velocity and the layer separation distance, respectively.

Similar to the traditional approach employed for the transport of natural particles (Soulsby, 1997), the fate of the deposited

particles is determined based on the magnitude of the bottom shear stress relative to critical values:

– if $|\tau_0| < |\tau_{cr,1}|$, the particle remains settled at the bottom;





- if $|\tau_{cr,1}| < |\tau_0| < |\tau_{cr,2}|$, the particle is transported as bedload (see Section 2.2.3)

- if $|\tau_0 > |\tau_{cr,2}|$, the particle is resuspended, allowing it to move again following the currents from its position on the bed.

These critical values, which can be either set up by users or calculated using empirical formulations, are written in terms of
critical Shield parameter, $\theta_c r$, as:

$$\tau_{cr} = \theta_{cr}(\rho_p - \rho)gD_p \tag{4}$$

where $g$ is the acceleration of gravity [m/s$^2$], $\rho_p$ the particle density [kg/m$^3$], $\rho$ the water density [kg/m$^3$], and $D_{eq}$ the equivalent particle diameter [-] calculated as

$$D_{eq} = \sqrt[3]{abc} \tag{5}$$

where $a$, $b$, and $c$ are the particle length [m], width [m] and height [m]. In TrackMPD v2.3, the model includes the modified Shield approach by Soulsby (1997), which provides the critical Shield parameters [-] for natural particles, for bedload

$$\theta_{cr,1} = \frac{0.3}{1 + 1.2D_*} + 0.055[1 - exp(-0.02D_*)] \tag{6}$$

and resuspension

$$\theta_{cr,2} = \frac{0.3}{1 + D_*} + 0.1[1 - exp(-0.05D_*)] \tag{7}$$

where $D_*$ [-] is the reduced diameter of MP particles. Additionally, the model incorporates the empirical formulation of Waldschläger and Schtrumpf (2019b), which accounts for the "hiding-exposure" effect of sediments and estimates the critical shear stress of microplastics deposited on sandy beds from the sediment shear stress as:

$$\theta_{cr,p}^* = 0.5588\theta_{cr}^*(\frac{D_{50}}{D_{eq}})^{-0.503} \tag{8}$$

where $\theta_{cr,p}^*$ is the critical Shields parameter of the microplastic [-], $\theta_{cr,sed}$ is the critical Shields parameter of the sediment
bed [-] calculated using equations (6) or (7), $D_{eq}$ [m] is the microplastic diameter and $D_{50}$ [m] is the median grain size of the sediment bed. In this study, we used Soulsby's formulations to calculate the critical shear stress of resuspension and bedload transport rather than Waldschläger's approach as the flume experiments were performed in the absence of sediments on the bed.

### 2.2.3   Bedload transport

When the bedload transport condition is satisfied, it is assumed that the near-bed particle is mainly driven by the drag force,
which is estimated as





$$F_d = \rho C_h U_r^2 D^2 \tag{9}$$

where $U_r$ the relative fluid velocity [m/s], i.e. the difference between the fluid and the particle velocities, $D$ a typical dimension of the particle [m] and $C_h$ is an empirical drag coefficient [-]. Vertical forces (lift and gravity/buoyancy) are expected to weakly affect the horizontal force balance, while friction, added mass and Basset forces are neglected but may be included in further developments of the model. Introducing the bed shear stress $\tau_0 = \rho u_*^2 = \rho C_d U^2$ [N/m$^2$], the hydrodynamic force can be written as:

$$F_d = \frac{C_h}{C_d} \frac{U_r^2}{U^2} \tau_0 D^2 \tag{10}$$

with $C_d$ the bottom drag coefficient [-]. Scaling the particle volume as $D_{eq}^3$ and using Equation 3, the particle acceleration can be written as

$$A = \frac{\nu}{D_{eq} \Delta z} \frac{\rho}{\rho_p} \frac{C_h}{C_d} \frac{U_r^2}{U} \tag{11}$$

Note that $\nu$ is here used instead of $\nu_t$ based on the assumption that turbulent mixing remains weak in the bedload-active layer of the flow. Finally, discretizing in time, the bedload particle velocity $U_b$ [m/s] at iteration $n+1$ can be written as

$$U_b^{n+1} = U_b^n + A\Delta t = U_b^n + \frac{\rho}{\rho_p} \frac{C_h}{C_d} \frac{\nu}{D_{eq}} \frac{U_r^2}{U} \frac{\Delta t}{\Delta z} \tag{12}$$

### 2.3 Experimental test case

The laboratory experiments conducted by Forsberg et al. (2020) in the CASH wind-wave flume (SEATECH/MIO) were reproduced using the SWASH-TrackMPD approach. The model domain was designed to replicate the configuration of the flume, which has a length of 6 m. The still water depth was set to 0.22 m, and a linearly sloping (1:20) bathymetry was implemented starting from x=1 m (Figure 1.a). The computational grid consisted of 175 points in the horizontal direction (resolution 3.45 cm) and 15 sigma-layers in the vertical direction. A time step of 0.05 seconds was selected for both hydrodynamic and particle tracking simulations. The SWASH model is forced at the left boundary (wave maker) by regular waves (period 1.2 s) associated with a weakly reflective boundary condition, while a Sommerfeld condition is applied at the right boundary (beach). The wave height at the boundary has been adapted to match the experimental measurement ($H = 9.2$ cm) at the first gauge ($X = 1.8$ m, see Fig. 1.a). While the model tends to underestimate the surf zone dissipation when compared to the experimental one, the comparison between experimental and numerical wave heights in Figure 1.a shows a correct overall agreement, allowing to use SWASH simulations as a relevant hydrodynamical forcing for TrackMPD.

In line with the flume experiments, six different types of MP are implemented in TrackMPD (Table 1). The particles were selected in Forsberg et al. (2020) to represent the range of shapes (including nearly spherical pellets, fibres, and sheets) and densities typically found in coastal waters. Table 1 provides a summary of the physical properties of the plastic particles, categorized as either light or heavy based on their densities relative to the water density of 1 g/cm$^3$. To accurately replicate



**Figure 1.** (a) Numerical domain replicating the experimental setup from Forsberg et al. (2019). The dashed black and blue lines represent the bottom depth and mean water level of still water, respectively. Numerical and experimental wave heights are depicted in black line and red circles, respectively. (b) Spatial evolution of the numerical wave skewness and asymmetry, in solid and dashed lines, respectively. (c) Contours of time-averaged horizontal velocity profile. (d) Contours of time-average vertical diffusivity coefficient for particle $K_v$ assumed to be equal to the eddy diffusivity $\nu_t$ from equation (1).





| Density classification | Shape | Density [g/cm$^3$] | Length [mm] | Thickness [mm] | Rising/Settling velocity [mm/s] |
|---|---|---|---|---|---|
| | Sphere | 0.92 | 3 | - | 77.5 |
| Low-density | Sheet | 0.92 | 5 | 0.1 | 3.9 |
| | Fiber | 0.95 | 5 | 0.5 | 4.7 |
| | Sphere | 1.38 | 3 | - | 110 |
| High-density | Sheet | 1.38 | 5 | 0.1 | 25 |
| | Fiber | 1.15 | 5 | 0.5 | 37.5 |

**Table 1.** Properties of MP released during the simulations.

205 the dynamic behavior of each particle, the model incorporated the measured rising or settling velocity values for each particle. These velocity values were obtained through precise measurements conducted in the laboratory using a vertical column with a specially designed gate at the bottom, following the protocol described in Jalón-Rojas et al. (2022).

The release of particles in the model accurately reproduced the conditions of the laboratory experiment. A total of 50 particles of each type (300 in total) were released at the beginning of the shoaling zone, at a depth between $z_r = 0.05$ and 0.15 m and 210 between $x_r = 1$ and 1.5m (see Section 2.4 for sensitivity tests on the release location). Each experiment ran for 10 minutes, allowing sufficient time for the system to reach a stationary state. To ensure the consistency and reproducibility of the model results, multiple runs of each reference scenario were performed and compared.

## 2.4 Sensitivity analysis

To evaluate the performance and sensitivity of the SWASH-TrackMPD model approach, we conducted a series of simulations 215 based on 6 Reference scenarios and 18 sensitivity scenarios (Table 2), keeping the hydrodynamics constant. The Reference scenario involved the modelling of the six types of particles presented in Table 1 while sensitivity scenarios are focused here on fibers, high and low density (similar trends have been observed for sheets and spheres).

Since the particle sizes and rising/settling velocities are known, the TrackMPD parameterization required only four free parameters: the horizontal and vertical diffusivity coefficients, $K_h$ and $K_v$, and the drag coefficients $C_h$ and $C_d$ used for bedload 220 transport calculations (Eq. 12). For each free parameter, representative values from the flume experiments were selected for the reference scenario (Table 2). A scaling factor of approximately 80 w.r.t real-scale environments was considered to estimate the diffusivity parameters. For example, the typical value of $10^{-4}$ m$^2$/s for $K_v$ in turbulent environments was transformed to 1.25 $10^{-6}$ m$^2$/s at the flume scale and selected as the background value (at the offshore and beach regions). The depth-dependent scaling presented in Section 2.2.1 is used to determine $K_v$ in the surf zone resulting in the cross-shore and vertically-varying 225 structure displayed in Figure 1.d. A common $K_h$ value in coastal environments (0.002/0.15 m$^2$/s at the flume/real scale) (Bogucki et al., 2005; Diez et al., 2008) was selected for the reference scenario. Different empirical drag coefficients for the





| Scenario | $K_h$ [m2/s] | $K_v$ [m2/s] | $C_D$ [-] | $C_H$ [-] | $x_r$ [m] | $z_r$ [m] | Tested parameter | Particles |
|---|---|---|---|---|---|---|---|---|
| Reference | $2 \times 10^{-3}$ | $1.25 \times 10^{-6}$ | 0.003 | 0.9 *spheres* <br> 0.8 *fibers* <br> 0.7 *sheets* | 1.5 | -0.1 | - | *spheres* <br> *fibers* <br> *sheets* |
| S1 | **0** | $1.25 \times 10^{-6}$ | 0.003 | 0.8 *fibers* | 1.5 | -0.1 | $K_h$ | *fibers* |
| S2 | $\mathbf{2 \times 10^{-2}}$ | $1.25 \times 10^{-6}$ | 0.003 | 0.8 *fibers* | 1.5 | -0.1 | $K_h$ | *fibers* |
| S3 | $2 \times 10^{-3}$ | $\mathbf{1.25 \times 10^{-5}}$ | 0.003 | 0.8 *fibers* | 1.5 | -0.1 | Offshore $K_v$ | *fibers* |
| S4 | $2 \times 10^{-3}$ | $\mathbf{1.25 \times 10^{-7}}$ | 0.003 | 0.8 *fibers* | 1.5 | -0.1 | Offshore $K_v$ | *fibers* |
| S5 | $2 \times 10^{-3}$ | $\mathbf{1.25 \times 10^{-5}}$ | 0.003 | 0.8 *fibers* | 1.5 | -0.1 | Uniform $K_v$ | *fibers* |
| S6 | $2 \times 10^{-3}$ | $\mathbf{1.25 \times 10^{-6}}$ | 0.003 | 0.8 *fibers* | 1.5 | -0.1 | Uniform $K_v$ | *fibers* |
| S7 | $2 \times 10^{-3}$ | $\mathbf{1.25 \times 10^{-7}}$ | 0.003 | 0.8 *fibers* | 1.5 | -0.1 | Uniform $K_v$ | *fibers* |
| S8 | $2 \times 10^{-3}$ | $1.25 \times 10^{-6}$ | **0.002** | 0.8 *fibers* | 1.5 | -0.1 | $C_d$ | *fibers* |
| S9 | $2 \times 10^{-3}$ | $1.25 \times 10^{-6}$ | 0.003 | **1 fibers** | 1.5 | -0.1 | $C_h$ | *fibers* |
| S10 | $2 \times 10^{-3}$ | $1.25 \times 10^{-6}$ | 0.003 | **0.4 fibers** | 1.5 | -0.1 | $C_h$ | *fibers* |
| S11 | $2 \times 10^{-3}$ | $1.25 \times 10^{-6}$ | 0.003 | 0.8 *fibers* | 1.5 | -0.1 | No bedload | *fibers* |
| S12 | $2 \times 10^{-3}$ | $1.25 \times 10^{-6}$ | 0.003 | 0.8 *fibers* | 1.5 | -0.1 | No resuspension | *fibers* |
| S13 | $2 \times 10^{-3}$ | $1.25 \times 10^{-6}$ | 0.003 | 0.8 *fibers* | 1.5 | -0.1 | Decrease $\tau_{cr2}$ | *fibers* |
| S14 | $2 \times 10^{-3}$ | $1.25 \times 10^{-6}$ | 0.003 | 0.8 *fibers* | 1.5 | -0.1 | Increase $\tau_{cr2}$ | *fibers* |
| S15 | $2 \times 10^{-3}$ | $1.25 \times 10^{-6}$ | 0.003 | 0.8 *fibers* | **1.3** | -0.1 | $x_r$ | *fibers* |
| S16 | $2 \times 10^{-3}$ | $1.25 \times 10^{-6}$ | 0.003 | 0.8 *fibers* | **1.1** | -0.1 | $x_r$ | *fibers* |
| S17 | $2 \times 10^{-3}$ | $1.25 \times 10^{-6}$ | 0.003 | 0.8 *fibers* | 1.5 | **0** | $z_r$ | *fibers* |
| S18 | $2 \times 10^{-3}$ | $1.25 \times 10^{-6}$ | 0.003 | 0.8 *fibers* | 1.5 | **-0.15** | $z_r$ | *fibers* |

**Table 2.** List of the tested scenarios in TrackMPD

particle bedload transport $C_h$ were selected as a function of the shape: 0.9 for sphere, 0.8 for fibers and 0.7 for sheets. A constant and uniform smooth bottom drag coefficient $C_d = 0.003$ is used for each simulation.

A series of sensitivity tests was then performed by varying individual parameters to assess their impact on particle dynamics
(Table 2). Scenarios S1 and S2 were designed to evaluate the sensitivity of particle dynamics to the horizontal turbulent diffusivity. Scenarios S3 and S4 examined the impact of modification of the offshore background $K_v$ while S5 to S7 explored configurations with uniform $K_v$ throughout the domain, with different tested values. S8 focused on the role of bottom drag. S9 and S10 explored the impact of increasing/decreasing the particle drag coefficient $C_h$. S11 to S14 focused on assessing the sensitivity of particle transport to the bottom dynamics condition. These scenarios included deactivation of bedload transport
(S11), deactivation of resuspension (S12), decrease the resuspension critical shear stress to the bedload value (S13) and increase the resuspension critical shear stress of one order of magnitude (S14). Additionally, scenarios S15 to S18 investigated the sensitivity of particle dynamics to different release points.





## 3 Results

### 3.1 Reference case

Figure 2 displays the comparison between Reference simulations and laboratory observations, for each type of particle. The results are interpreted in terms of trajectories throughout the whole simulation, along with their final positions, and cross-shore MP distributions in five distinct regions: offshore, shoaling, breaking, surf, and beach zones. Overall, the simulations demonstrate a satisfactory reproduction of the spatial distribution of the different particle types, providing a first demonstration of the reliability and accuracy of the SWASH-TrackMPD approach in capturing the dynamics of microplastics in nearshore

waters. Notably, to ensure the robustness of our simulations, we conducted five simulations for each scenario, and the results consistently exhibited only minor variability in the number of particles within each region. This variability was within the same order of magnitude as that observed in the experiments as indicated by the error bars in Figure 2.

Consistent with the experimental observations, the cross-shore distribution of the low-density MP varied depending on their shape, highlighting the diverse range of dynamic behaviors exhibited by light particles (Fig. 2.a-c). Low-density spheres,

characterized by strong rising velocities, were primarily transported onshore in the surface layer through the action of Stokes drift and wave non-linearities as analysed in Section 3.3. Consequently, all the particles eventually reached the beach (Fig. 2.a). While also experiencing an overall onshore motion, low-density sheets depicted a broader distribution throughout the water column and in the final cross-shore position (Fig. 2.b). Low-density sheets also exhibited higher variability in the final number of particles within each region in both experiments and simulations. These results can be attributed to their enhanced

mobility across different layers of the water column, which are influenced by distinct transport mechanisms as further discussed in Section 3.3. Like the flume observations, the model predicted the highest number of low-density sheets finishing in the surf region at the end of the experiment (Fig 2.b). However, it exhibited a marginal overestimation in the number reaching the beach compared to the observed data. While difficult to interpret with the present data, this small difference may be attributed to inaccuracy in the beach zone definition and/or to complex small-scale processes related to sheet particle beaching not accurately

represented by the present transport approach. Fibers show an intermediate transport pattern between spheres and sheets. Finally, 54% and 67% of low-density fibers were gradually transported onshore and eventually reached the beach in the simulations and observations, respectively. A small portion of particles displayed a wider distribution in the final cross-shore position due to their mobility within the water column (Fig. 2.c).

The simulated high-density particles moved onshore very close to the bed in accordance with the flume observations, yet

their final position was slightly offshore. In contrast to the experimental results where particles were predominantly trapped in the breaking region, in the model, they tended to accumulate in the upper shoaling region (Fig. 2.d-f). These particles settled rapidly upon release and gradually migrated onshore due to the non-linear wave behavior in the shoaling zone, until reaching a region where they remained relatively stationary, as further elaborated in Section 3.3. Discrepancies in the accumulation region between simulations and observations may be attributed to slight misrepresentation of the experimental undertow current or the

temporal evolution of vertical mixing by the model, or inaccuracies in the definition of the compartment limits.





*(left and right panels, rows (a)–(f))*

**Figure 2.** Trajectories (left panel) and final cross-shore distribution (right panel) of the 6 plastic particles with indications of the location of the offshore, shoaling, breaking, surf and beach zone: (a) low-density spheres; (b) low-density sheets; (c) low-density fibers; (d) high-density spheres; (e) high-density sheets; (f) high-density fibers. Right panels compare the results of the experimental observations Forsberg et al. (2020) (red bars) and the simulations from this work (blue bars). The error bars represent the standard deviation from two experiments and five simulations. Red crosses, grey lines and black dots in left panels represent the release points, the trajectories and the final position of particles.





## 3.2 Sensitivity Analysis

This section presents the results of the sensitivity analysis, aiming to assess the influence of model parameters that were not experimentally quantified on particle dynamics. This analysis also seeks to gain a better understanding of the relative significance of key processes, which include enhanced turbulence induced by wave breaking, bedload transport, and resuspension.

Figure 3 illustrates the impact of varying the horizontal ($K_h$) and vertical ($K_v$) diffusivity parameters on the final cross-shore distribution of low and high-density fibers. When we decreased $K_h$ to 0 m$^2$/s, we observed minor changes in the transport trends and final distribution of both low-density and high-density particles (Fig. 3.b). When $K_h$ was increased to 0.02 m$^2$/s (i.e., 10 times the Reference value), it had little impact on low-density particle trends (Fig.3.i.c), but an alteration occurred for high-density particles (Fig. 3.ii.c). This increase in $K_h$ hindered approximately one-third of the high-density fibers from moving

onshore, while facilitating another third to reach the breaking zone. This outcome is likely due to an excessive increase in the stochastic transport component in the horizontal direction.

Variations in $K_v$ also had minimal effects on the transport patterns of low-density particles, which primarily undergo suspension transport. As depicted in Figure 3.i.d-g, the cross-shore distribution of low-density fibers remained consistent across multiple scenarios: (a) the Reference case, where $K_v$ exhibited spatial variability with higher values in the breaking zone (Fig.

3.i.a); (b) when $K_v$ was reduced or increased by a factor of 10 in the offshore region (S3-S4, Fig. 3.i.d and Fig. 3.i.e); (c) when $K_v$ had constant values (ranging from 1.25x10$^{-5}$ to 1.25x10$^{-7}$ m$^2$/s, S5-S7) throughout the entire domain (Fig. 3.i.f and Fig. 3.i.g). On the other hand, the transport behavior of high-density particles is not very sensible to the background $K_v$ at the offshore region (S3-S4, Fig. 3.ii.d-e) but is affected when considering constant turbulent conditions through the whole domain (S5-S6). As shown in Fig. 3.ii.f high-density fibers failed to move onshore when constant $K_v$ dropped below 10$^{-5}$ m$^2$/s (Fig.

3.ii.f and Fig. 3.ii.g). This phenomenon is primarily attributed to the contribution of eddy diffusivity, $\nu_t$, on the bed shear stress leading to potential resuspension rather than a potential effect on the stochastic particles' dispersion through the water column. When $\nu_t$ reached relatively low values in the shoaling region, the decrease of shear stress (eq. (3)) precludes the activation of resuspension condition (Section 2.2.2) preventing particle resuspension and subsequent onshore transport.

These findings emphasize the significance of accounting for enhanced mixing resulting from wave breaking, as well as the

importance of accurately estimating the order of magnitude for $K_v$ and $K_h$ to faithfully reproduce the transport of high-density microplastics in nearshore environments but allowing for some uncertainty in their values. Therefore, the Feddersen formulation offers a physically-sound estimate of the order of magnitude for $\nu_t/K_v$ and a more realistic depiction of their spatial evolution, without the necessity of using overestimated and/or empirical values through the whole domain.

The modification of particle and bottom drag coefficients does not appear to affect the cross-shore distribution of particles

(Fig. 4a-d). The effect of bedload transport (comparison between Figs. 4.a and .e) appears also to be weak in the present configuration. A much stronger effect is observed for resuspension (comparison between Figs. 4.a and .f). When resuspension is totally deactivated, high-density particles were not able to shift onshore. One notes that the modification of the resuspension threshold, either decreasing (Fig. 4.g) or increasing (Fig. 4.h), does not affect the main dynamics. This absence of sensitivity, which indicates that the resuspension is overwhelmed in typical conditions, strengthens the finding of a primary role played by





resuspension in the transport of heavy particles. Finally, the results remained virtually insensitive to variations in the release
point's position (S15-S18, not shown). Only the release of particles closer to the offshore regions (S16) resulted in a small
portion of particles remaining in the offshore region.

### 3.3 Dominant transport mechanisms

In this Section, we examine the core physical processes that govern the transport of microplastics in the simulated scenario. As
analyzed in Section 3.1, distinct transport trends were evident for low-density and high-density microplastics. To explore these
divergent behaviors, Figures 5 and 6 depict the temporal evolution of the vertical and horizontal trajectory of a low-density sphere
and a high-density fiber as they traverse the shoaling and breaking regions. These figures also include the temporal evolution of
water level ($\eta$), current magnitude ($|U|$), and current direction ($U_{Dir}$) at the particles' positions. Complementary information
on the cross-shore distribution of wave and wave-driven parameters, such as the mean wave height $H$, wave skewness $Sk$ and
asymmetry $As$ (calculated following Grasso et al. (2011)), and residual currents $U_{res}$, is provided in Figure 1.

Low-density particles exhibited net onshore transport in the horizontal dimension (Fig. 2.a) driven by the Stokes drift and
wave asymmetry. As shown in Figure 5.a-f, these spheres predominantly traveled in the upper water layer, closely following the
water surface in the vertical coordinate, as their high buoyancy prevents significant dispersion due to turbulence. Consequently,
they followed the net drift velocity aligned with the wave propagation direction, commonly known as the Stokes drift (van den
Bremer and Breivik, 2018) (see residual velocity in Figure 1.c). The overall transport mechanism is therefore that plastic particles
undergo stronger shoreward motion at greater height reached under the crest than under the trough. This residual onshore
transport is quite marginal near the offshore region (Fig. 5.c) in comparison with the upper shoaling zone or the breaking region
(Fig. 5.d). Figure 5.h highlights that currents responsible for transporting these particles exhibit significantly higher magnitudes
around the crest when directed onshore ($U_{Dir}$ close to 0°in Fig. 5.j), compared to the offshore direction ($U_{Dir}$ close to ±180°).
This asymmetry between onshore and offshore transports became more pronounced within the breaking zone, characterized by
prolonged shoreward motion ($-90 > U_{Dir} < 90°$), as depicted in Figure 5.f. This observation finds further support in the strong
negative asymmetry combined with positive skewness observed within this region, as illustrated in 1.b.

Low-density sheets and fibers had higher vertical mobility (Fig. 2.c and Fig. 2.e) and were therefore influenced by different
transport mechanisms: vertical turbulent mixing, Stokes drift and onshore residual currents enhanced by wave non-linearity
at surface layers, and offshore undertow currents at bottom layers (Fig. 1.b-c). As discussed by Forsberg et al. (2020), the
sensitivity of the position of these particles along their trajectory can be compared with the uncertainty observed in deterministic
chaos systems, which results in a broader spatial dispersion of particles.

Figure 6.i illustrates the vertical behaviour of the three types of high-density particles and the underlying transport processes
using fibers as an example. These particles were typically deposited at the bottom by gravity forces, frequently coinciding with
the transition between onshore and offshore current phases and vice-versa, when velocities were at their minimum. Resuspension
typically occurred after deposition, causing particles to make small jumps in both onshore and offshore directions. As depicted
in Figure 6.c, these jumps led to a minimal residual onshore transport, while accumulation after the passage of multiple waves
resulted in the transport of particles toward the upper shoaling region. Resuspension events, which can also be interpreted as



**Figure 3.** Final cross-shore distribution of low-density (left pannels) and high-density (right pannes) fibers with indications of the location of the offshore, shoaling, breaking, surf and beach zone for different scenarios: (a-b) Reference, $K_h$=6.3x10$^{-3}$, varying $K_v$, $K_v, off$=1.25x10$^{-4}$; (c-d) S1, $K_h$=1.25x10$^{-2}$; varying $K_v$, $K_v, off$=1.25x10$^{-4}$; (e-f) S2, $K_h$=6.3x10$^{-2}$; varying $K_v$, $K_v, off$=1.25x10$^{-4}$; (g-h) S3, $K_h$=6.3x10$^{-3}$; varying $K_v$, $K_v, off$=1.25x10$^{-5}$; (i-j) S4, $K_h$=6.3x10$^{-3}$; varying $K_v$, $K_v, off$=1.25x10$^{-6}$; (k-l) S5, $K_h$=6.3x10$^{-3}$; uniform $K_v$, $K_v$=1.25x10$^{-4}$; (m-n) S7, $K_h$=6.3x10$^{-3}$; uniform $K_v$, $K_v, off$=1.25x10$^{-8}$. The red and blue bars represent the results of the experimental observations from Forsberg et al. (2020) and the simulations from this work, respectively. The error bars represent the standard deviation from two experiments.







**Figure 4.** Final cross-shore distribution of high-density fibers with indications of the location of the offshore, shoaling, breaking, surf and beach zone for different scenarios: (a) Reference $C_d$=0.003, $C_m$=0.8; (b) S8, $C_d$=0.002; (c) S9, $C_m$=1; (d) S10, $C_m$=0.4; (e) S11, bedload deactivated; (f) S12, resuspension deactivated; (g) S13 $\tau_{cr2}$=$\tau_{cr1}$; (h) S14 $\tau_{cr2}$x10. The red and blue bars represent the results of the experimental observations from Forsberg et al. (2020) and the simulations from this work, respectively. The error bars represent the standard deviation from two experiments.





saltation transport, appear here to dominate over the pure bedload motion (rolling/sliding). While producing overall consistent

particle fluxes at the beach scale, no direct validation of the numerical near-bed dynamics can be performed using the present experimental data. As detailed in Section 3.2, resuspension could only take place if turbulence is high enough to facilitate it. Upon approaching the breaking region (Fig. 6.ii), high-density particles exhibited a similar vertical behavior, but the residual onshore transport was offset by stronger return currents (Fig. 1.c).

## 4    Discussion

The results of this study underscore the robustness of the proposed modeling approach in advancing our understanding of microplastic transport, reinforcing several consistent findings from prior research in this field, and introducing new insights, discussion points, and research perspectives. Previous studies have also recognized differences in the behavior between low-density and high-density particles and highlighted the key role of the settling/rising velocity (also expressed non-dimensionally as the Dean number) in the transport of plastic particles (Alsina et al., 2020; Kerpen et al., 2020; Guler et al., 2022; Larsen et al.,

2023; Núñez et al., 2023). The experimental results from Kerpen et al. (2020), Larsen et al. (2023), and Núñez et al. (2023) also demonstrated a net onshore transport and beaching of very low-density particles. Much like our study, Kerpen et al. (2020) and Larsen et al. (2023) illustrated that highly buoyant particles are transported at higher mean velocities and are more likely to beach due to their greater tendency to remain near the surface and escape from undertow return flow. On the other hand, Alsina et al. (2020) conducted an in-depth examination of the role of Stokes drift in the transport of plastic particles, but unlike our

study, it focused solely on shoaling waters, omitting an examination of the effect of the non-linearity of breaking waves. Also in line with our work, Kim and Kim (2024) emphasized the significance of wave breaking on MP transport. They demonstrated that buoyant particles with more neutral buoyancy are particularly influenced by undertow currents, resulting in delayed beaching compared to high-buoyant ones.

     While recent research has advanced the understanding of wave non-linear processes (e.g. Martins et al., 2020), their effects on

plastic transport have received limited attention. The findings of our study suggest that the primary mechanism for high-density microplastic transport is linked to wave asymmetry, similar to sand transport dynamics. Grasso et al. (2011) demonstrated that wave asymmetry promotes the resuspension of sand particles during the rising phase of the wave crest's velocity, followed by sedimentation, resulting in onshore transport during the waning phase until the return undertow flow counteracts this transport. This cycle is less apparent in our results, likely due to the constrained temporal evolution of vertical turbulence. Nonetheless, we

also observed residual onshore transport following the passage of multiple waves. Future research may involve a more robust parametrization of this parameter to better capture its effect on particle transport. Furthermore, Guler et al. (2022) investigated the cross-shore distribution of various non-buoyant particles under different beach configurations, namely a plane bed and a barred beach. They observed varying transport patterns dependent on the particle's Dean number and the beach morphology. For instance, non-buoyant particles with a high Dean number (indicating relatively lower settling velocity) tended to migrate to the

beach region on the plane bed, while a hotspot of particles formed on the plateau of the barred beach. The authors attributed





**Figure 5.** Temporal evolution of the depth of a low-density sphere during its transit through the shoaling (a) and breaking (b) regions, accompanied by corresponding time series of environmental hydrodynamic variables at the particle's location: water level elevation (c-d), current velocity (e-f), and velocity direction.





**Figure 6.** Temporal evolution of the depth of a high-density fiber during its transit through the shoaling (a) and breaking (b) regions, accompanied by corresponding time series of environmental hydrodynamic variables at the particle's location: water level elevation (c-d), current velocity (e-f), and velocity direction.





these differences to wave skewness and asymmetry, which increased in the surf zone of the plane bed configuration, facilitating onshore particle transport, while decreasing in the plateau region of the barred beach configuration, hindering onshore migration.

Núñez et al. (2023) also evaluated the transport behavior of high-density microplastics through laboratory experiments, and observed that they were mainly trapped in the breaking zone. They also found that the cross-shore distribution of particles
remained consistent under both regular and irregular wave conditions, with variations in cross-shore transport processes linked to the different time scales associated with these conditions. It should be highlighted that, in addition to the physical properties of the particles, beach morphology and wave forcing appear to play crucial roles in determining transport trends, although further investigation is needed. Wind conditions (Forsberg et al., 2020) and longitudinal currents may also impact particle transport. Consequently, the interpretation of the present results and previous research needs to be framed in the context of the specific
experimental configurations employed, which are inherently limited in experimental studies. For example, the experiments from Forsberg et al. (2020) simulated in this work are characterized by stormy conditions, including relatively high and non-linear waves, as well as the presence of intense turbulent conditions right from the onset of the shoaling region due to the wave-maker action. Less energetic conditions may thus lead to different transport patterns. The robustness and consistency demonstrated by our numerical model offer a valuable opportunity to continue building knowledge on these aspects through holistic studies
that evaluate the influence of various geomorphological and wave climate conditions, in addition to exploring the underlying physical processes.

Future modelling studies using the proposed SWASH-TrackMPD approach also offer a promising avenue to bridge the gap between laboratory experiments and the complexities of real-world nearshore environments. In experimental studies, scaling effects present a significant challenge when attempting to accurately replicate the complexities of natural systems within the
controlled laboratory setting. While laboratory experiments involve scaling down physical parameters such as wave heights and velocities to fit the laboratory's constraints, microplastic properties such as density, and therefore rising/settling velocities, are typically left unscaled. As discussed above, this (unscaled) parameter plays a pivotal role in predicting microplastic transport trends and hotspot formation. Our sensitivity analysis (Section 3.2) even demonstrated that this parameter is the primary particle property driving its transport, as the choice of erosion threshold did not significantly impact transport patterns under the
simulated conditions. Future modelling studies using real-scale morphologies, environmental forcings, and accurate particle properties will be essential for validating these significant results and for reevaluating microplastic transport trends discussed in previous experimental research. These studies will also contribute to a deeper understanding of the physical processes governing transport, as discussed in this study.

The SWASH-TrackMPD modelling approach stands out as quite unique in its methodology and offers several distinct
advantages and advanced features in (a) resolving wave transformation dynamics; (b) coupling with surf-zone turbulence parameterization, or potentially with any other turbulent modelling, for accurate prediction of wave-induced turbulence effects; (c) simulating resuspension and bedload processes, drawing from traditional sediment research (Soulsby, 1997) and pioneering microplastic studies (Waldschläger and Schttrumpf, 2019b); (d) implementing advanced settling and rising velocity formulations, including biofouling processes (Jalón-Rojas et al., 2022; Baudena et al., 2023). The validation provided in this work is also
unique in the literature. Several simplifications have been used to build the transport model in a computationally-efficient





perspective. The estimation of bed shear stress for resuspension and bed load transport was based on the molecular viscosity, i.e. assuming that turbulent mixing does not play a significant role in the near-bed region. Further research works at finer scale are required to better frame this assumption in realistic conditions in terms of particle size, bed roughness and wave boundary layer dynamics. In addition, it is worthwhile to note that the particle near-bed dynamics, which has been shown here to be

dominated by resuspension/saltation events rather than pure sliding/rolling bedload processes, can not be directly validated due to the lack of relevant measurements. Further detailed tracking of wave-driven near-bed transport regimes should be envisioned by future laboratory works to provide suitable validation data for wave-resolving transport models. More generally, advanced turbulence models able to inject, at the wave scale, the breaking-induced mixing in the turbulence parameterization should bring finer insight on the time-resolved particle dispersion throughout the water column. The model also neglects non-inertial effects

induced by particle properties on advection. This omission is supported by the experimental findings of Alsina et al. (2020), which indicated that, apart from the buoyancy, such properties exert minimal influence on the net drift of low-density particles in the shoaling region. The authors also suggested that the net drift of high-density particles might be influenced by plastic density and size, but trends remain inconclusive due to particle motion variability. Indeed, in turbulent environments, the pronounced stochastic transport characteristics may outweigh the influence of non-inertial particle effects on advection. Nevertheless, future

model developments could consider incorporating the effects of particle drag on advection, as proposed by Stocchino et al. (2019), to further investigate these hypotheses and refine our understanding of microplastic transport dynamics in nearshore environments.

## 5   Conclusions

The SWASH-TrackMPD modelling approach has proven to be a pertinent, robust and versatile tool for investigating microplastic

transport in nearshore environments. It offers unique features in terms of its ability to resolve wave transformation dynamics, consider wave-breaking-induced turbulence, and simulate resuspension and bedload processes tailored for diverse types of plastic particles. These advancements have culminated in the release of TrackMPD v3.0, marking a significant milestone in our ability to simulate microplastic dynamics in coastal environments.

   The model reproduced the wave laboratory experiments conducted by Forsberg et al. (2020), accurately simulating the

distribution of particles of different densities, sizes and shapes over the beach profile and providing new insights into the key mechanisms governing their movement. Our findings underscore the critical role of rising and settling velocities in the transport of microplastics. Low-density microplastics exhibited net onshore transport driven primarily by the Stokes drift and wave asymmetry. While highly buoyant particles remained predominantly near the water surface with the net drift aligned with wave propagation direction, particles characterized by lower rising velocities were more affected by turbulent motion and had a

broader final distribution over the system.

   High-density microplastics exhibited a distinct transport pattern, predominantly trapped near the breaking zone. Akin to sediment transport dynamics, the breaking zone accumulation tends to result from the competing effects of near-bed transport driven by wave asymmetry and return undertow flow.



The consistency observed in the results of the sensitivity analysis also underscores the reliability of the model. It highlights
the importance of accurately estimating the order of magnitude for diffusivity parameters, particularly for vertical turbulence, to
ensure a faithful representation of high-density microplastic transport while allowing for some degree of uncertainty in value
estimation. The adoption of the Feddersen formulation provides a physically sound estimate for these parameters and a more
realistic description of their spatial distribution. Furthermore, this sensitivity analysis reaffirms the primary role played by
resuspension in the transport of high-density particles, while revealing that the selection of the bottom drag coefficients barely
affects the outcomes.

The SWASH-TrackMPD modelling approach emerges as a valuable tool for continuing the investigation of plastic particle
migration and accumulation in the nearshore, effectively bridging the gap between laboratory experiments and real-world coastal
dynamics. We recommend future modelling studies with a similar approach to the present study but incorporate real-scale
morphologies and environmental forcings. Extending this modelling framework into three dimensions will enable a more
comprehensive exploration of the impact of various environmental factors, including wind conditions, longitudinal currents,
and rip currents. These future modelling implementations hold the potential to significantly enhance our understanding of the
nearshore's dual role as a source and sink of plastics, improve the parameterization of beaching and refloating in global numerical
models, and contribute to the development of more effective management and mitigation strategies for plastic pollution in
coastal regions.

*Code availability.* Software name: TrackMPD (https://doi.org/10.5281/zenodo.12514513)

Developers: Isabel Jalón-Rojas and Vincent Marieu (from v2); Isabel Jalón-Rojas, Erik Fredj and Xiao Hua Wang (v1).

Contact information: isabel.jalon-rojas@u-bordeaux.fr

Year first available: 2019 (v1); 2023 (v2); 2024 (v3)

Hardware requirements: PC, HPC

System requirements: Linux, Windows

Program language: Matlab

Program size: 1.83 Go

Availability: https://github.com/IJalonRojas/TrackMPD

License: GNU Public License (GLP-3.0)

Documentation: README and manual in the Github repository.

*Author contributions.* **I.Jalón-Rojas**: Conceptualization, Methodology, Software, Formal Analysis, Validation, Writing - Original Draft,
Writing - Review and Editing. **D. Sous**: Conceptualization, Methodology, Validation, Writing - Review and Editing. **V. Marieu**: Methodology,
Software, Validation, Writing - Review and Editing.





*Acknowledgements.* This work was supported by the French national program EC2CO (Ecospheere Continentale et Cotiere) through the project PLASTICBEACH.





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
