# Peer review of "A wave-resolving 2DV Lagrangian approach to model microplastic transport in the nearshore based on TrackMPD 3.0"

_Geoscientific Model Development, 2024_

## Author Response (AR1)

**We thank the editor and both reviewers for their careful reading of our manuscript, their constructive and positive comments, and valuable suggestions. Below, we address each point raised and outline the changes made in response.**

Editor

Please note that for your paper, the following requirements have not been met in the Discussions paper:

- "The main paper must give the model name and version number (or other unique identifier) in the title."

In order to simplify reference to your developments, please add a model name (and/or its acronym) and a version number in the title of your article in your revised submission to GMD.

**Following the journal's requirements, we have modified the title to include the model name and version number as 'A wave-resolving 2DV Lagrangian approach to model microplastic transport in the nearshore based on TrackMPD v3.0''**

Reviewer 1

Minor comments

L131-138: In Section 2.2.1, general aspects and model improvements are described, without addressing the specific configuration for the present study. Therefore, I would suggest relocating these lines (L131-137, "For the present simulation…. uniform throughout the domain.") closer to line L223, where the chosen model parameters for the study are detailed.

**We have modified this part of the text to clarify the general aspect applicable to other study cases in the nearshore (lines 133-140):** *"An additional background eddy viscosity can be imposed at the beach and offshore regions. At the beach zone, the Feddersen scaling is not applicable due to the impossibility to compute energy flux in the intermittently wet and dry portion of the beach. A smoothing is applied to avoid sharp vt gradients in transition areas.*

*We propose that, for nearshore applications, as in the present study, the vertical diffusivity coefficient for particles, Kv , instead of being constant, can be assumed to be equal to the eddy diffusivity calculated using this approach, meaning that momentum and plastic particles diffuse at the same rate. This assumption is reasonable, given that the small size of microplastics makes their dynamics to be largely dominated by turbulence in these highly dynamic systems, but it requires further validation through dedicated research efforts.."*

**We have also moved specific information to the present study to Section 2.4 as suggested by the reviewer :**

**(lines 230-232):** *"For example, the typical value of 10−4 m2/s for Kv in turbulent environments was transformed to 1.25 10−6 m2/s at the flume scale and selected as the*

*background value (at the offshore, to account for turbulence generated in the vicinity of the wave maker, and beach regions).***”**

**(lines 233-234) “***The horizontal diffusivity coefficient for particles is here kept uniform throughout the domain.***”**

L133: Please replace "surfzone" with "surf zone".

**Done.**

L136: Here, a sentence could be added to justify the hypothesis that the vertical diffusion coefficient for microplastics equals the eddy diffusivity, given that the small size of microplastics ensures their behaviour as passive tracers predominantly governed by fluid turbulence.

**Good suggestion, we have included this explanation in the text as suggested (lines 138-140): "This assumption is reasonable given that the small size of microplastics *makes their dynamics to be largely dominated by turbulence in these highly dynamic systems, but it requires further validation through dedicated research efforts.***".**

L190: I would explicitly state that one of the two windless hydrodynamic conditions evaluated by Forsberg was replicated numerically.

**This precision has been added in the text (line 193-194): "***The laboratory experiments conducted by Forsberg et al. (2020) in the CASH wind-wave flume (SEATECH/MIO) were reproduced using the SWASH-TrackMPD approach, specifically under one of the two windless hydrodynamic scenarios***."**

L199: It is recommended to include a quantitative/statistical comparison between numerical and laboratory wave heights and an assessment of how much the model underestimates dissipation in the surf zone.

**We have modified the sentence to include a quantitative comparison (lines 203-207): "Numerical wave heights are generally in good agreement with experimental measurements, as illustrated in Figure 1a. Specifically, the model tends to slightly underestimate surf zone dissipation, leading to a root mean square error (RMSE) of 2.2 cm and an average negative bias of 0.7 cm for wave heights in this region. Despite this minor discrepancy, the overall agreement in wave heights supports using SWASH simulations as a reliable hydrodynamic forcing for TrackMPD."**

L207: I believe the Jalon-Rojas method (2022) estimates rising and settling velocities in calm water. De Leo et al. (2021; https://doi.org/10.3390/jmse9020142) found that settling velocities increase in the presence of waves. I would add a sentence noting this aspect.

**Yes, as discussed in more detail in the response to Reviewer 2, our approach neglects particle inertia, and this should indeed be more explicitly stated and discussed. We have expanded on this by further elaborating the discussion, which already referenced studies such as Alsina et al. (2020), suggesting that in the nearshore shoaling region—our area of interest—inertial effects minimally influence the net drift of low-density particles, aside from buoyancy. Additional references have been added to strengthen the discussion and highlight potential future developments, including the mentioned study in lines 53 and 456.**

**Please refer to the response to Reviewer 2 for a complete description of the changes made on this aspect.**

L212 & L245: It should be clarified whether the results presented in the manuscript are the average of all runs or from a representative run.

**Figures 2.b and 3.a-b illustrate the main results of the manuscript, showing the average values and standard deviations from five runs for each scenario. As noted in the text: "to ensure the robustness of our simulations, we conducted five simulations for each scenario, and the results consistently exhibited only minor variability in the number of particles within each region. This variability was within the same order of magnitude as that observed in the experiments as indicated by the error bars in Figure 2." In Figures 2.a and 3.c, we present the trajectories and final positions from one of the runs, which, given the low variability between runs, is representative of the scenario. This last point has been clarified in the figure caption :**

**"*In the left panels, red crosses, grey lines and black dots represent the release points, the trajectories and the final position of particles for one of the five simulations, representative of the scenario.*"**

P11, Section 3 (Results): Similarly to line L261, where it states "...54% and 67% of low-density fibres were gradually transported onshore...", I would like to see more quantification in the description of spheres and sheets throughout this section.

**For low-density spheres and all high-density particles, nearly all particles ended up in a single compartment, as already mentioned in the text (lines 260 and 276). We have included this quantification for low-density sheets as suggested by the reviewer (lines 266-268): "*However, it slightly underestimated the number of particles in this region (40% on average in simulations compared to 60% in observations) while overestimating the number reaching the beach (24% in simulations versus 9% in observations).*".**

Reviewer 2

General comments

Modelling the transport of plastic debris is known to be a challenging task for the complex mechanisms involved. The inertial character of the debris poses the major problems and several (simplified) solutions have been proposed in the last years. TrackMPD is one of the

available suite for modelling the Lagrangian transport of particles in ocean and coastal environments. A major concern remains how the inertial transport is modelled.

Focusing the attention on the hydrodynamic context described in the manuscript, a series of papers have been recently published on the transport of inertial particles under the action of waves. For examples the following contributions:

DiBenedetto, M. H., Ouellette, N. T., & Koseff, J. R. (2018). Transport of anisotropic particles under waves. Journal of Fluid Mechanics, 837, 320-340.

DiBenedetto, M. H., & Ouellette, N. T. (2018). Preferential orientation of spheroidal particles in wavy flow. Journal of Fluid Mechanics, 856, 850-869.

DiBenedetto, M. H., Koseff, J. R., & Ouellette, N. T. (2019). Orientation dynamics of nonspherical particles under surface gravity waves. Physical Review Fluids, 4(3), 034301.

De Leo, A., & Stocchino, A. (2022). Dispersion of heavy particles under sea waves. Physics of Fluids, 34(1).

DiBenedetto, M. H., Clark, L. K., & Pujara, N. (2022). Enhanced settling and dispersion of inertial particles in surface waves. Journal of Fluid Mechanics, 936, A38.

Clark, L. K., DiBenedetto, M. H., Ouellette, N. T., & Koseff, J. R. (2023). Dispersion of finite-size, non-spherical particles by waves and currents. Journal of Fluid Mechanics, 954, A3.

The above list is not complete, but represent a good example on how the inertial behaviour of the plastic debris can be mathematically modelled. The main conclusions of the studies on inertial particles and waves are: settling/rising is strongly enhanced by the inertial effects and the spreading on inertial particles is almost never found to be fully diffusive.

It is clear that it is almost impossible to fully include the inertial effects in approaches like the ones used in TrackMPD. However the Authors should include at least part of the reference listed above and discuss the limitations of their modelling approach. For example, the assumption that the dispersion of the plastic debris is diffusive should be better explained. Closing the fluxes using a diffusion-type coefficient is a common approach even if it has been demonstrated that heavy particles don't follow a Brownian dispersion regime. However, it is convenient to introduce horizontal and vertical coefficient, as done in the present study. TrackMPD seems to include only gravitational effects through a settling velocity of the debris. No other inertial effect are included or modelled. The Authors should discuss these aspects in more details.

**We thank the reviewer for the valuable comment and the list of publications on the inertial behavior of particles. It is true that our approach neglects particle inertia, and this should indeed be more explicitly stated and discussed. In our previous version, our discussion already touched on this point, where we reference the experimental study by Alsina et al. (2020), which suggests that, in the nearshore shoaling region—our area of interest—factors aside from buoyancy, such as inertial effects, minimally influence the net drift of low-density particles, likely due to high turbulence and other complex processes in this region (air-water two-phase mixing, roller). We also note that future**

**model developments could consider incorporating the effects of particle drag on advection, as proposed by Stocchino et al. We agree that the studies cited by the reviewer provide valuable insights into particle inertia, and we have expanded our discussion in the revised version of the manuscript using these references to inform potential future developments (lines 447-457):**

*"While a wide range of research has highlighted the importance of inertial effects on particle advection and settling under wave oscillatory flow (DiBenedetto et al., 2018, 2019, 2022; De Leo et al., 2021; De Leo and Stocchino, 2022), this omission is supported in nearshore waters by the experimental findings of Alsina et al. (2020), which indicated that, apart from the buoyancy, such properties exert minimal influence on the net drift of low-density particles in the shoaling region. The authors also suggested that the net drift of high-density particles might be influenced by plastic density and size, but trends remain inconclusive due to particle motion variability. Indeed, in this region characterized by strong turbulent conditions and other complex processes (air-water two-phase mixing, roller), the pronounced stochastic transport characteristics may outweigh the influence of non-inertial particle effects on advection. Nevertheless, future model developments could consider incorporating the effects of particle drag on advection and settling, as proposed by Stocchino et al. (2019) and De Leo et al. (2021), to further investigate these hypotheses and refine our understanding of microplastic transport dynamics in nearshore environments."*

Another major concern regards the treatment of the particle transport close to the bed (sections 2.2.2 and 2.2.3). To what extent is reasonable to apply concepts developed for sediment transport to the present case? the empirical formulations used in the case of bedload or suspended load are developed considering relatively high concentration of sediment, especially in the case of bedload. with this assumption, the emprical models are not lagrangian models, but describe the average behavior of a certain mass of sediment. Indeed, the sediment continuity equation (Exner equation) is written in Eulerian terms. Similarly, suspended sediment transport is described using the advection diffusion equation for the sediment concentration. Plastic debris concentrations are usually much less compared to suspended sediment concentration and, of course, at the bed. The Authors used the Soulsby formulation designed for sands. Is it reasonable to assume that the estimate of the critical shear stresses (bedload and suspension) is valid also in case of plastic debris? There are no evidence of this in the literature, even if this approach is commonly used. Moreover, it is not clear the formulation of the bedload transport and how, provided the mobility condition, the bedload is described in terms of an acceleration term.

**We agree that there are some differences, but also similitudes, in the transport of sediment and microplastics as has been reviewed in the work by Waldschlager et al. (2022). We also concur with the reviewer that there is a lack of empirical work on microplastics, particularly for bed load transport. Therefore, we made the assumption of using a similar approach to that used for sediments as an initial step, which can be refined in the future as more empirical data become available.**

**Regarding the calculation of the bed shear stress of microplastics, there are now several empirical studies available. For example, Waldschlager and Schttrumpf (2019) proposed a correction of Shield formulation to account for the sediment bed properties. Goral et al. (2023) proposed a new framework so that the incipient motion conditions for microplastic particles lying on a sediment bed are, for the first time, reconciled with the classical Shields diagram. One author of the present manuscript is also working on this**

topic (https://meetingorganizer.copernicus.org/EGU24/EGU24-20316.html). **Given the current state of knowledge, TrackMPD include the Soulsby/Shield approach and the correcting proposed by Waldschlager and Schttrumpf (2019). Given that there is no sediment bed in the experimental case that we are reproducing, we have opted for using the Soulsby approach for this specific case. The approaches included in TrackMPD can be refined in future as more empirical data will be available. We will discuss these elements in more detail in the revised manuscript, addressing both the current limitations and the potential for future refinements based on emerging empirical data (lines 424-433):**

**"*The estimation of bed shear stress for resuspension and bed load transport was based on the molecular viscosity, i.e. assuming that turbulent mixing does not play a significant role in the near-bed region, with the critical values for motion initiation derived from formulations for natural particles. While using the Soulsby formulation for sediments is reasonably valid for spherical particles, it may be less accurate for fibers and sheets due to their anisotropic shapes and complex interactions with the flow. Even though our sensitivity test suggested that this parameter had a secondary effect on the transport of high-density microplastics in the flume, future studies will incorporate new empirical formulations for the incipient motion of microplastics with different shapes as they become available.*"**

**Waldschläger, K., Brückner, M. Z., Almroth, B. C., Hackney, C. R., Adyel, T. M., Alimi, O. S., ... & Wu, N. (2022). Learning from natural sediments to tackle microplastics challenges: A multidisciplinary perspective. *Earth-Science Reviews*, *228*, 104021.**

**Waldschläger, K. and Schttrumpf, H.: Erosion behavior of different microplastic particles in comparison to natural sediments, Environmental science & technology, 53, 13 219–13 227, 2019b.**

**Goral, K. D., Guler, H. G., Larsen, B. E., Carstensen, S., Christensen, E. D., Kerpen, N. B., ... & Fuhrman, D. R. (2023). Shields diagram and the incipient motion of microplastic particles. Environmental Science & Technology, 57(25), 9362-9375.**

Detailed comments

Introduction

Include at least part of the reference provided above and discuss in more details what have been done in order to describe the inertial particles transport under waves

**We have completed the description of the state of the art using references provided above (lines 50-54): "To our knowledge, prior to this study, Stocchino et al. (2019) and De Leo and Sottochino (2020) have conducted such a fine numerical studies evaluating the effects of sea waves on inertial microplastic dynamics, a topic that has also been well documented through experimental research (DiBenedetto et al., 2018, 2019, 2022; De Leo et al., 2021).**

Section 2 Methods

1. Please provide more details on the Lagrangian particle transport model equation implemented in TrackMPD

**The advection-dispersion equations and numerical schemes implemented in TrackMPD were comprehensively detailed in the original publication (Jalón-Rojas et al., 2019). In this paper, we focus on presenting the new developments of the model rather than repeating previously published information. However, we do provide a summary of the key aspects in lines 91-93: "*The model employs a 4th-order Runge-Kutta scheme to accurately advect virtual particles through a set of velocity fields. A random-walk approach is implemented to simulate the turbulent motion of particles in both the horizontal and vertical directions as a function of the horizontal and vertical diffusivity coefficients (see Section 2.2.1 for more details).*" For more comprehensive information on the Lagrangian transport equations, we direct interested readers to the earlier publication (Jalón-Rojas et al., 2019).**

2. Please provide more details on the mesh used in SWASH and the integration time step used for both SWASH and TrackMPD.

**The mesh and time steps used in SWASH and TrackMPD were already provided in lines 196-198: "*The computational grid consisted of 175 points in the horizontal direction (resolution 3.45cm) and 15 sigma-layers in the vertical direction. A time step of 0.05 seconds was selected for both hydrodynamic and particle tracking simulations.*"**

3. What are the Stokes time of the simulated particles? is it comparable to the integration time step?

**The integration time step (0.05 seconds) is much smaller than the Stokes time for all particles (ranging from 9.4 to 35.7 seconds, calculated as $\tau\_Stokes=d_p^2/(12v\beta)$), indicating that the model's time step is adequately small to resolve the dynamics of the particles accurately. This text has been added in lines 198-200.**

4. line 165, please provide the definition of D*

**Done.**

5. Section 2.2.3: what is the difference between Ch and Cd?

**Ch refers to the particle drag coefficient, while Cd refers to the bottom drag coefficient. We have completed the description of Ch in line 181 to avoid any confusion between the two terms.**

6. line 185. It is not clear the reason why the eddy viscosity is replaced the fluid viscosity

**The following explanation has been included in the Discussion section (lines 433-437):**

**"*Regarding the assumption of molecular viscosity, estimating the Stokes layer thickness as $\sqrt{2v/\sigma}$ for the present case leads to a value about 0.6mm, which is larger than the thickness of tested fibers and sheets. For the spheres, the top of the particles is expected to rise above the laminar layer. However, as the exact distribution of small-scale*"**

*viscous/turbulent shear remains unaccessible to the present dataset, the fluid viscosity is used for each particle type to ensure robust comparisons.*"

7. Section 3.1. The results shown in Figure 2 are not clear. How particles have been simulated to obtain the results? Are the grey lines the particle trajectories? If yes, why the black points represent the final positions? How long was the simulations in terms of wave periods? and what is the influence on the final results?

**We have clarified the color coding (lines 249-251): "The results are interpreted in terms of trajectories throughout the whole simulation (grey lines), along with the final positions (black dots), and cross-shore MP distributions in five distinct regions (color bars)".**

**The simulations run for a sufficiently long duration (10 minutes, compared to the 1.2-second wave period) to ensure that the results are not influenced. We have included this information in the revised version lines (217-219). Sensitivity tests were even conducted to confirm this.**

8. section 3.3. The discussion should be improved providing more details on the physical meaning of the transport mechanisms

**We have added more discussion elements on the physical meaning of the transport mechanism:**

**(lines 327-333): "superimposed to the oscillating backwards and forwards transport due to the wave motion, the plastic particles tend to spend more time in the faster onshore-moving layer underneath the crest than in the slower offshore-moving layer below the trough. This effect, which is the well-known Stokes drift, is enhanced by the increased non-linearity and asymmetry of shoaling and surf zone waves. As shown in Figure 5.a-f, these spheres predominantly traveled in the upper water layer, closely following the water surface in the vertical coordinate, as their high buoyancy prevents significant dispersion due to turbulence. Consequently, they followed the net drift velocity aligned with the wave propagation direction, i.e. the Stokes drift (van den Bremer and Breivik, 2018) (see residual velocity in Figure 1.c)."**

**(lines 342-345): "In surface layer, the driving mechanisms are the same that affected low-density particles described before. When spreading throughout the water column due buoyancy-driven settling and vertical turbulent mixing, the particles are exposed to a offshore-directed return current, i.e. the classical undertow, which compensates for the wave-induced surface mass flux (Fig. 1.b-c)."**

9. line 360. it is not true that the interaction between plastic particles and waves received limited attention, see the reference listed above. Please rephrase

**We agree that the interactions between plastic particles and waves have not received limited attention. However, in this sentence in question, we specifically refer to wave**

**non-linear processes (e.g., wave asymmetry and skewness) in shallow waters, which have received less attention. To better clarify it, we have added "in shallow waters" after "wave non-linear processes (e.g. Martins et al., 2020)" (line 376).**

10 a section on the limitation of the approach should be added

**The main limitations of our approach and potential future improvements were already included in the discussion section. Nevertheless, we have divided the Discusion in two subsections with the second one focusing on limitations and perspectives. The text in this subsection has been completed by including some points raised in general comments to ensure that the limitations (and future developments) are more explicitly detailed (lines 426-457).**

---

## Author Response (AR2)

**We thank the reviewer for this last minor comment, which is addressed below:**

L326-347. For the low-density particles, there is a residual onshore transport, which may be due to Stokes drift in the offshore zone and also wave asymmetry in the shoaling zone, but in the breaking zone this transport is primarily due to the undertow profile. This should be clarified in the text of the manuscript.

**We have modified the text to clarify this point as (L341-348):**

**"Low-density sheets and fibers had higher vertical mobility (Fig. 2.c and Fig. 2.e) and were therefore influenced by different transport mechanisms. In surface layer, the driving mechanisms for a residual onshore transport are the same that affected low-density particles described before. When spreading throughout the water column due buoyancy-driven settling and vertical turbulent mixing, the particles are exposed to a offshore-directed return current, i.e. the classical undertow, which compensates for the wave-induced surface mass flux (Fig. 1.b-c). This undertow-driven transport is expected to be particularly effective in the breaking zone. As discussed by Forsberg et al. (2020), the sensitivity of the position of these particles along their trajectory can be compared with the uncertainty observed in deterministic chaos systems, which results in a broader spatial dispersion of particles."**